# HARNESSING THE POWER OF LARGE LANGUAGE MODELS FOR NATURAL LANGUAGE TO FIRST-ORDER LOGIC TRANSLATION

## ABSTRACT

Translating natural language sentences to first-order logic (NL-FOL translation) remains a critical task in many logic-based NLP systems, as it enables ML models to reason logically over text. However, existing translation methods still struggle to scale to real-world tasks due to the lack of a large and high-quality dataset and a model family with high precision and coverage.

In this work, we approach this longstanding challenge by harnessing the power of pre-trained large language models (LLMs). To do so, we present MALLS (large language Model generAted NL-FOL pairS), a dataset of 28K diverse and verified sentence-level NL-FOL pairs collected from GPT-4. We create MALLS by implementing an adaptive pipeline that prompts GPT-4 for pairs with rich and diverse contexts. To ensure the validity of FOL rules and their alignment with the NL sentences, we utilized a combined strategy of FOL rule parsing, human annotation, and automatic filtering.

We also present LOGICLLAMA, a LLaMA2-7B/13B model family fine-tuned on MALLS for NL-FOL translation. LOGICLLAMA is capable of directly translating natural language into FOL rules, which outperforms GPT-3.5. LOGICLLAMA is also equipped to correct FOL rules predicted by GPT-3.5, and can achieve similar performance as GPT-4 with a fraction of the cost. This correction ability was achieved by a novel reinforcement learning with human feedback (RLHF) framework, which initially trains on synthetically perturbed NL-FOL pairs to encourage iterative correction and then fine-tunes with RLHF on GPT-3.5 outputs using a FOL verifier as the reward model. Codes and data are available here.

## 1 INTRODUCTION

Natural language to first-order logic (NL-FOL) translation is a critical task that serves as the foundation of a wide range of logic-backed NLP applications, such as textual entailment (Bos & Markert, 2005; Han et al., 2022; Pan et al., 2023), NL inference (Angeli & Manning, 2014), theorem proving (Polu & Sutskever, 2020; Abzianidze, 2017), and question answering (Wang et al., 2021). For example, in a textual entailment task, one is given a set of NL premises and is tasked to predict the label of an NL hypothesis. A logic-based method approaches this by first translating the NL premises and hypothesis into FOL facts and rules and then solving the logic system with logic programming tools such as Prolog or SMT solvers. Traditionally, NL-FOL translation has been addressed via rule-based methods (Abzianidze, 2017; Zettlemoyer & Collins, 2005; Bos & Markert, 2005). Due to the complexity of natural language, these methods are difficult to scale to real-world applications, and the lack of a reliable translation model remains the main bottleneck for these systems to generalize to more tasks: if one can reliably extract the formal system from the raw text, then the rest can be solved with standard logic programming.

The recent remarkable progress of large language models (LLMs) gives rise to a new paradigm: using LLMs to perform the bulk of the translation task, thereby benefiting from their generalization capabilities and capacity to handle complex and diverse language constructs. One straightforward approach to harness the power of LLMs for NL-FOL translation is via in-context learning with powerful models such as GPT-3.5 and GPT-4. While this approach has seen success in this task (Pan

et al., 2023), it is closed-source and lacks extendability for downstream tasks (more discussion in §2). More importantly, we show in §3 that even the most powerful GPT-4 can still fail significantly on simple tasks and exhibit biased behaviors.

In this work, we adopt the self-instruct and fine-tune paradigm (Wang et al., 2023) used for many general-purpose LLMs such as Alpaca (Taori et al., 2023) and Vicuna (Chiang et al., 2023), where we collect and process data from GPT-4 and then fine-tune open-source LLMs (i.e., LLaMA) on it.

To fine-tune LLMs for translation, one needs high-quality NL-FOL pairs. Existing datasets such as LogicNLI (Tian et al., 2021) and FOLIO (Han et al., 2022) are either synthetic or too small for sufficient fine-tuning. We present MALLS, a 28K real-world, diverse, and verified sentence-level NL-FOL pairs collected from GPT-4. To create MALLS, we implement an adaptive pipeline that prompts GPT-4 for pairs with rich and diverse context. Knowing that GPT-4 can generate misaligned FOL rules, we take extra caution in verifying the generated pairs. We verify the dataset with human annotation, identify categorical mistakes made by the GPT-4, and filter the dataset from 34k to 28k. We further analyze the type of errors GPT-4 made, revealing categorical areas of weakness in the latest LLMs. Compared to the existing datasets (Table 1), MALLS enjoys better diversity in terms of vocabulary, context, and complexity, and is the largest dataset to date in this category.

On top of MALLS, we present LogicLLaMA, a LLaMA-7B/13B model family (Touvron et al., 2023) for NL-FOL translation fine-tuned with LoRA (Hu et al., 2021). LogicLLaMA can be used for (1) directly translating NL to FOL as a standalone translator and (2) can be used in combination with more powerful general-purpose models such as GPT-3.5, where it serves a "correction model" that corrects outputs from GPT-3.5, which we found yields better performance with a fraction of the cost of GPT-4 [1]. In particular, we propose a novel reinforcement learning with human feedback (RLHF) framework that first trains LogicLLaMA on the synthetically perturbed NL-FOL pairs, equipping LogicLLaMA with generating corrective prompts, and then fine-tunes it with RLHF on the GPT-3.5 outputs using a FOL verifier as the reward model.

We summarize our contributions as follows: (1) We present MALLS, a 28K real-world, diverse, and verified sentence-level NL-FOL pairs collected from GPT-4, which, to the best of our knowledge, is the largest dataset to date in this category. (2) We also present LogicLLaMA, a LLaMA-7B/13B model family fine-tuned on MALLS, which, in the experiments, achieves GPT-4 level performance on NL-FOL translation tasks on three benchmarks.

## 2 RELATED WORK

**NL-FOL translation**. Traditionally, NL-FOL translation has been addressed via rule-based methods (Abzianidze, 2017; Zettlemoyer & Collins, 2005; Bos & Markert, 2005). Due to the complexity of natural language, these methods are difficult to scale to real-world applications. Recently, there has been an increasing interest in approaching this task via neural approaches (Lu et al., 2022; Cao et al., 2019; Hahn et al., 2022; Wang et al., 2021; Singh et al., 2020; Levkovskyi & Li, 2021), which gives rise to a new paradigm of using LLMs for translation. Recent works such as Logic-LM(Pan et al., 2023) have successfully used GPT-3.5 for translation via in-context learning. While the GPT-based approach demonstrates impressive few-shot capabilities in this task, it comes with several limitations: (1) As of Sept, GPT-4 costs $0.06/1K tokens and can be expensive to run; GPT-3.5, while costs less, struggles with complex translation (§5); (2) they are closed-source and subject to constant updates, making it difficult for privacy-sensitive use cases and reproducing results for academic purposes; (3) they lack extendability for future integration with downstream tasks such as NLI and QA. Therefore, is it valuable to have small and open-source LLMs with GPT-level performance while preserving privacy and extendability. And in this work, we investigate this paradigm and propose to collect NL-FOL pairs from GPT-4 and fine-tune LLaMA-7B/13B on it.

**NL-FOL datasets**. Many datasets that focus on logical reasoning ability have been proposed recently. For example, LogiQA (Liu et al., 2020), RuleTaker (Clark et al., 2020), ReClor (Yu et al., 2020) and text2log (Levkovskyi & Li, 2021). However, these datasets either do not provide sentence-level FOL annotations, or the annotations are generated without verification. Among these works, LogicNLI (Tian et al., 2021) and FOLIO (Han et al., 2022) are closest to our work, which provides NL statements with parallel FOL annotations. However, pairs in LogicNLI are generated synthetically

---

[1]As of Sep 2023, GPT-3.5 costs $0.002/1K tokens for completion whereas GPT-4 costs $0.06/1K.

and share a similar FOL template. FOLIO consists of real-world expert-written pairs, but the size of 2K is insufficient for fine-tuning an LLM. This work extends the prior work and proposes to collect NL-FOL pairs from GPT-4. As a result, MALLS has collected 28K pairs that are more diverse in terms of context and complexity. In experiments, we evaluate LOGICLLAMA on MALLS as well as on LogicNLI and FOLIO and demonstrate that MALLS is of high quality and enables models to generalize to other benchmarks as well.

## 3 MALLS DATASET CREATION

We create the MALLS dataset by collecting NL-FOL pairs from GPT-4 which is considered to be the most powerful LLM to date. As of Sept 2023, MALLS has reached the size of 28K and we plan to continue expanding the dataset in future versions.

### 3.1 PROMPT PIPELINE

To collect data from GPT-4, we implemented a prompting pipeline that dynamically adjusts the prompts to both ensure the *diversity* and *validity* of the NL-FOL pairs. The pipeline consists of the following modules: (1) **N-gram frequency counter**; (2) **Prompter**; and (3) **FOL rule verifier**.

**N-gram frequency counter**. During prompting, we keep track of the frequencies of the N-grams in the entire NL statement corpus. Specifically, we track 1- and 3-grams. Once the frequency of a specific N-gram in the collected data reaches the frequency threshold (500 and 250 respectively), we will instruct GPT-4 to not produce any NL-FOL pairs including it. For example, "... *DO NOT involve concepts and terms (and the synonyms) such as animal, food, ...*". The list of N-grams in the instruction grows as more reach the frequency threshold.

**Prompter**. A prompter assembles the prompts generated from different modules (prompt table shown in Appendix B): (1) SYSTEM PROMPT: specifying the basic requirements such as the syntax and generation format. (2) FEW-SHOT EXAMPLES PROMPT: consisting 5 NL-FOL pair examples randomly sampled from the corpus. Initially, pairs are sampled from the FOLIO dataset and later on from the GPT-4-generated ones (we checked to ensure none of the FOLIO examples, or close variations are leaked into the GPT-4 generated NL-FOL pairs.). This diversifies the prompts and leads to less similar examples .(3) NEGATIVE N-GRAM PROMPT: instructing GPT-4 not to involve frequent N-grams (introduced earlier) in the generated NL-FOL pairs. (4)FOL PROMPTS: generating prompts that specify the desired form of FOL rules, i.e., the number of variables and whether or not to include more logic operators such as $\oplus$, $\neg$, and $\vee$ which we found GPT-4 tends to ignore in default generation. These configurations are picked randomly every time the prompt is generated. (5) BREAK-DOWN PROMPT: We found GPT-4 by default tends to make over-complicated predicates that absorb important logical meanings. For example, " ### NL: *A fruit is considered ripe if it is mature and its color has changed from green to red.* ### FOL: $\forall x(\texttt{Fruit}(x) \land \texttt{Mature}(x) \land \texttt{ColorChangedToRed}(x) \rightarrow \texttt{Ripe}(x))$. " The predicate $\texttt{ColorChangedToRed}$ is complicated and should be broken down into "$\texttt{ColorBefore}(x,y) \land \texttt{ColorAfter}(x,z) \land \texttt{Green}(y) \land \texttt{Red}(z)$". We detect long predicate names and include a prompt encouraging the model to break down the rules.

**FOL rule verifier**. GPT-4 can sometimes generate syntactically invalid FOL rules. We implement a verifier that checks the syntax of the rules. Specifically, we specify the context-free grammar (CFG) of the expected FOL rule and parse the generated FOL with NLTK [2] CFG parser, and erase those that could not be parsed (grammar and example parse trees in Appendix B).

### 3.2 NL-FOL ALIGNMENT CHECK

Apart from the validity of the FOL rules, it is also critical to ensure FOL rules align with the NL sentences. The straightforward approach is to check the entire set manually as in the creation of prior datasets such as FOLIO (Han et al., 2022). However, conducting a full annotation is prohibitive for the size of MALLS (x14 times larger) for an academic budget. Instead, we employ a hybrid approach where we first manually annotate a subset of MALLS, identify the common error modes, and then implement a filtering module to filter those in the entire set. We find this approach to be sufficiently effective and increase the correct ratio from 84.5% to 92.8%.

---

[2] https://www.nltk.org/

Table 1: Statistics of MALLS, LogicNLI, and FOLIO datasets.

| Dataset | Source | #NL-FOL pairs | NL | | | FOL | | | | | | | |
|---------|--------|---------------|----|----|----|-----|----|----|----|----|----|----|----|
| | | | Vocab size | Avg. #words | Avg. #literals | $\forall$ | $\exists$ | $\neg$ | $\wedge$ | $\vee$ | $\rightarrow$ | $\leftrightarrow$ | $\oplus$ |
| FOLIO[3] | Expert | 2K | 5105 | 10.4 | 2.1 | 1111 | 182 | 421 | 631 | 167 | 1137 | 17 | 121 |
| LogicNLI[3] | Synthetic | 12K | 2061 | 13.9 | 2.8 | 2783 | 5327 | 10230 | 6590 | 2373 | 8712 | 3288 | 0 |
| MALLS | GPT-4 | 28K | 20959 | 15.7 | 4.6 | 27256 | 1738 | 3443 | 25209 | 5164 | 25212 | 1835 | 1777 |

MALLS **Annotation**. We annotate a subset of 1K examples uniformly sampled from MALLS. Each sample is rated with respect to a 3-point scale: correct, partial, and incorrect. Each sample is evaluated by 3 annotators and gets the final rating via majority votes. We recruited 6 graduate students in CS/ECE department with a background in FOL. The annotators were provided with detailed instructions regarding the rating scales, and several examples per rating to clarify further (details in Appendix B). The annotators were given 3 hours to finish the rating of 500 examples. The labeled 1K subset consists of 84.5% correct, 11.8% partial, and 3.7% incorrect pairs.

**Filtering** MALLS. We analyze the 1K samples, identify four common error modes, and design corresponding modules to filter them: (1) Free variables. GPT-4 occasionally generates FOL rules with free variables such as "$\forall x P(x) \wedge R(x, y)$", where $y$ is not bounded. We remove them by parsing the rule and checking if it contains free variables. (2) Nested equivalence/implication. For some complex NL statements such as "*If an A has B, C, D then it is E*", GPT-4 tends to confuse between "$\wedge$", "$\rightarrow$", and "$\leftrightarrow$" and generates inaccurate rules such as "$\forall x A(x) \rightarrow (B(x) \wedge C(x) \wedge D(x) \leftrightarrow E(x))$". We find samples with this nested structure all have a similar alignment issue, so we remove these pairs by parsing the rule and checking if it contains such a structure. (3) Aggressive xor. We find GPT-4 struggles with the meaning of "$\oplus$", and may generate a rule such as "$A(x) \oplus B(x)$" for an NL sentence "*A or B*". To remove those pairs with aggressive xor usage, we construct a 5-shot prompt and use GPT-3.5 to identify the potential pairs and then remove the incorrect ones manually. (4) Missing NL information. We find GPT-4 may omit important entities or concepts in the FOL rule. We adopt a similar treatment as (3), where we first do 5-shot in-context learning with GPT-3.5 to identify potentially incorrect samples and then remove the true incorrect ones manually. After removing samples of the four error modes, the resulting 1K samples have 92.8% correct, 6.0% partial, and 1.2% incorrect pairs. We filter the raw MALLS of 34K samples the same way and obtain the final 28K verified samples.

## 3.3 DATASET STATISTICS

**General statistics**. We show the general statistics in Table 1 together with those of LogicNLI and FOLIO[3]. MALLS contains 28K NL-FOL pairs, which is significantly larger than LogicNLI and FOLIO, and different from LogicNLI which is synthetically generated, the pairs are also more diverse and contextually rich, where the NL statements have a vocabulary size of 20.9K and an average length of 15.7 compared to 10 in FOLIO. For FOL rules, the average number of literals reached 4.6 indicating more complex rules (also see Figure 9 in Appendix B).

**Pair diversity**. The NL-FOL rules in MALLS are highly diverse. To see this, we investigate the frequencies and the correlations of the FOL *terms*. A *term* is either a predicate name or a named entity in a FOL rule. For example, "$\forall x((\texttt{Person}(x) \wedge \texttt{Drinks}(x)) \rightarrow \texttt{DependentOn}(x, \texttt{Caffeine}))$" consists of 4 terms, i.e., `Person`, `Drinks`, `DependentOn` and `Caffeine`. MALLS has a total term vocabulary size of 49394 and the most frequent terms occur less than 2K times (Figure 8 in Appendix B), suggesting a diverse vocabulary distribution. On the other hand, we investigate the correlations between terms and illustrate the top 200 frequent term pairs. We show a snippet of this in Figure 1 (for the full version, see Figure 7 in Appendix B). Note that if a term is associated with many other terms, this typically means the rules involving that term are diverse in semantics and context, and Figure 1 suggests that it is indeed the case. For example, for rules involving `Book`,

---

[3]Note that the FOLIO statistics are different from those reported in (Han et al., 2022). As of Sep 2023, the released dataset misses the ground truth FOL annotations for conclusions in the training set, and some pairs contain duplicates and invalid FOL rules. We removed those during pre-processing. Also, the LogicNLI statistics are obtained from the official repo here, which contains 12K samples instead of the 20K reported in the paper.

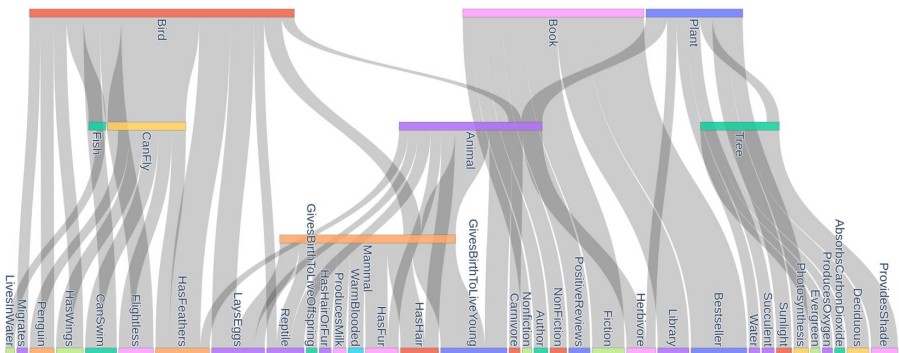

Figure 1: Snippet from the top 200 frequent FOL term pairs in MALLS (for full version see Appendix B). Many terms are associated with a wide range of other terms, which suggests the rules are semantically and contextually diverse.

Figure 2: Input and expected outputs for direct translation, naive correction, and iterative correction.

they cover the knowledge of its genre (e.g., `Fiction`), places (e.g., `Library`), viewership (e.g., `Bestseller` and `PositiveReviews`), and so on.

# 4  LOGICLLAMA FOR NL-FOL TRANSLATION

In this section, we discuss how to fine-tune the LLaMA-7B/13B (Touvron et al., 2023) model on the MALLS to reach a GPT-4 level performance, which we refer to as LOGICLLAMA. Unlike typical NLP tasks, where one fine-tunes it with a task-agnostic objective such as autoregression, fine-tuning for NL-FOL translation is nontrivial. Specifically, we address the following challenges: **(C1) How to fine-tune LOGICLLAMA for optimal performance?** In §4.1, we first consider the direct approach, where LOGICLLAMA is trained to predict the correct FOL in one go. In §4.2, we also propose an iterative approach, where LOGICLLAMA corrects the FOL produced by GPT-3.5 step by step via RLHF, which yields better performance. **(C2) How to evaluate the FOL rules?** Unlike NL text, evaluating an FOL rule against the ground truth is nontrivial as it requires parsing the rule and comparing the actual logical meanings. In §4.3, we propose two metrics to measure this.

## 4.1  FINE-TUNING FOR DIRECT TRANSLATION AND NAIVE CORRECTION

The LOGICLLAMA can be trained to directly translate the FOL from NL, which we refer to as **(T1) direct translation** task; it can also be trained to *correct* the generated FOL from a more powerful model such as GPT-3.5, which we refer to as the correction task. In this section, we consider the **(T2) naive correction** approach, where the correction is done in one go. The intuition is that we found in experiments GPT-3.5 is good at doing the "heavy-lifting" part of the translation and can capture the main part of the FOL rule; then presumably, one can train a smaller model that corrects the output from the GPT-3.5 to get a better result.

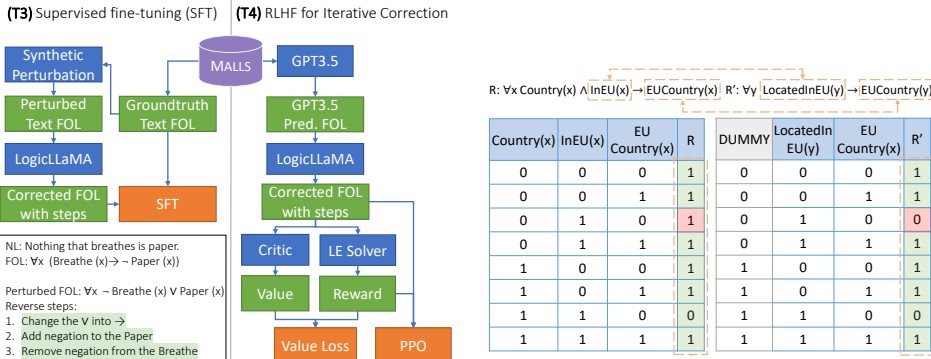

Figure 3: Overview of the SFT and RLHF training for the iterative correction.

Figure 4: Example of computing the Logical Equivalence score, 7/8=0.875.

We train both **(T1)** and **(T2)** via standard autoregression objective. Specifically, we fine-tune LLaMA-7B/13B with LoRA for all the attention and feedforward linear layers on MALLS. The left two columns in Figure 2 show the input and output sequence of the two tasks: let $\langle x_{\text{NL}}, x_{\text{FOL}} \rangle$ be an NL-FOL pair from MALLS; for **(T1)**, the input and output are the original sequences $x_{\text{NL}}$ and $x_{\text{FOL}}$ respectively; and for **(T2)**, let $\hat{x}_{\text{FOL}} = \text{GPT}(x_{\text{NL}})$ be the FOL predicted by GPT-3.5, the input is the NL and the prediction put together $[x_{\text{NL}}, \hat{x}_{\text{FOL}}]$ and the output is the ground-truth FOL, $x_{\text{FOL}}$.

## 4.2 ITERATIVE CORRECTION VIA RLHF

While **(T1)** direct translation and **(T2)** naive correction are easy to train, they do not lead to optimal performance. Inspired by the recent chain of thought technique (Pan et al., 2023; Wei et al., 2022), we found that training the model to produce the intermediate steps during the correction often leads to better performance. Such examples are shown in Figure 5. To train such a model, one needs a dataset consisting of not only the ground-truth $\langle x_{\text{NL}}, x_{\text{FOL}} \rangle$, but also the correction steps specific to a predicted FOL. Formally, recall that $\hat{x}_{\text{FOL}}$ is the predicted FOL by GPT-3.5, then we need the ground-truth steps $\hat{\mathcal{X}}_\Delta = [\hat{x}_{\Delta,1}, \hat{x}_{\Delta,2}, ..., \hat{x}_{\Delta,T}]$, such that they form a valid sequence $[\hat{x}_{\text{FOL}}, \hat{x}_{\Delta,1}, \hat{x}_{\Delta,2}, ..., \hat{x}_{\Delta,T}, x_{\text{FOL}}]$. The right column of Figure 2 shows a 4-step correction example.

However, as stated in **(C1)**, we do not have ground-truth steps $\hat{\mathcal{X}}_\Delta$ for the predicted FOL from GPT-3.5. We propose to address this issue using a combination of supervised fine-tuning (SFT) and RLHF training on the real GPT-3.5 output with a logical equivalence solver (discussed in §4.3) as the reward model. Specifically, we refer to the SFT step as **(T3) SFT Correction**. And as shown in the left column of Figure 3, we create a synthetic FOL dataset by perturbing the ground-truth FOL rule and obtaining the ground-truth steps by reversing the past perturbations. And, we refer to the RLHF step as **(T4) RLHF Correction**, which is shown in the right column of Figure 3.

**FOL Rule Perturbation and SFT correction**. To generate samples with $\hat{\mathcal{X}}_\Delta$, we generate synthetic steps by randomly perturbing the FOL rules in MALLS. We leave details of perturbation to Appendix C. We generate 150K perturbed samples in the form of $\langle x_{\text{NL}}, x_{\text{FOL}}, \hat{\mathcal{X}}_{\Delta,\text{prev}}, \hat{\mathcal{X}}_{\Delta,\text{corr}}, \hat{x}_{\text{FOL}} \rangle$, where $\hat{\mathcal{X}}_{\Delta,\text{prev}}, \hat{\mathcal{X}}_{\Delta,\text{corr}}, \hat{x}_{\text{FOL}}$ are the previous correction steps, target correction steps and the perturbed FOL rule respectively (example shown in Figure 2 right column). Note that here, $\hat{\mathcal{X}}_\Delta$ is effectively $\hat{\mathcal{X}}_{\Delta,\text{prev}} + \hat{\mathcal{X}}_{\Delta,\text{corr}}$. We split the steps into two parts so that the model learns to generate steps not only from scratch but also from previous corrections; this ability is later used in RLHF. Finally, for **(T3)** SFT correction, we fine-tune the LLaMA-7B/13B with LoRA again using the standard autoregression objective: the input is $[x_{\text{NL}}, \hat{x}_{\text{FOL}}, \hat{\mathcal{X}}_{\Delta,\text{prev}}]$ and the output is $[\hat{\mathcal{X}}_{\Delta,\text{corr}}, x_{\text{FOL}}]$.

**RLHF Correction**. With **(T3)** SFT correction, we enable the model to generate intermediate correction steps for synthetic data. Now, we train the model to correct the actual outputs from GPT-3.5, which is **(T4)** RLHF Correction task. Note that to achieve this goal for **(T4)**, we can no longer use the autoregression objective as in **(T1)**, **(T2)**, or **(T3)**, since we still do not have the ground-truth steps for GPT-3.5 outputs. However, on the other hand, we can still compare the final

corrected rule to the ground-truth rule and measure how close they are. And this gives rise to an RL approach to the problem. Formally, let $\text{RM} : \mathcal{X} \times \mathcal{X} \mapsto [0, 1]$ be a function that maps a pair of FOL sequences, $\boldsymbol{x}_{\text{FOL}}$ and $\boldsymbol{x}'_{\text{FOL}}$, to a scalar score representing the pair similarity, our objective can be formalized as maximizing the score (effectively the expected return in RL),

$$\max_\pi \text{RM}(\boldsymbol{x}_{\text{FOL}}, \boldsymbol{x}'_{\text{FOL}}), \text{ where } \boldsymbol{x}'_{\text{FOL}} \sim \pi_\theta(\boldsymbol{x}_{\text{FOL}}, \hat{\mathcal{X}}_{\Delta,\text{corr}} | \boldsymbol{x}_{\text{NL}}, \hat{\mathcal{X}}_{\Delta,\text{prev}}, \hat{\boldsymbol{x}}_{\text{FOL}}), \quad (1)$$

for all tuples $\langle \boldsymbol{x}_{\text{NL}}, \boldsymbol{x}_{\text{FOL}}, \hat{\boldsymbol{x}}_{\text{FOL}} \rangle$ in MALLS via a policy $\pi_\theta(\boldsymbol{x}_{\text{FOL}}, \hat{\mathcal{X}}_{\Delta,\text{corr}} | \boldsymbol{x}_{\text{NL}}, \hat{\mathcal{X}}_{\Delta,\text{prev}}, \hat{\boldsymbol{x}}_{\text{FOL}})$ which is exactly the autoregressive model we trained in **(T3)** and would like to fine-tune in **(T4)**. With objective Eq.(1), task **(T4)** is now similar to the RLHF[4] proposed in InstructGPT (Ouyang et al., 2022) with the only difference being the reward model RM, where in our case, RM is a logical equivalence solver (§4.3) instead of a language model.

**Training process**. In **(T4)** RLHF correction, we fine-tune the LOGICLLAMA model obtained in **(T3)** SFT correction via RLHF. For every tuple $\langle \boldsymbol{x}_{\text{NL}}, \boldsymbol{x}_{\text{FOL}}, \hat{\boldsymbol{x}}_{\text{FOL}} \rangle$, we let the model to continuously generate the corrections $[\langle \boldsymbol{x}'^{(1)}_{\text{FOL}}, \hat{\mathcal{X}}^{(1)}_{\Delta,\text{corr}} \rangle, \langle \boldsymbol{x}'^{(2)}_{\text{FOL}}, \hat{\mathcal{X}}^{(2)}_{\Delta,\text{corr}} \rangle, ...]$ until the model outputs "No changes needed" in the steps or hits the token limit; the previous correction $\hat{\mathcal{X}}_{\Delta,\text{prev}}$ is set to empty initially and we update it with the output steps in every generation. In other words, at iteration $(t)$, the previous correction is $\hat{\mathcal{X}}_{\Delta,\text{prev}} = [\hat{\mathcal{X}}^{(1)}_{\Delta,\text{corr}}, \hat{\mathcal{X}}^{(2)}_{\Delta,\text{corr}}, ..., \hat{\mathcal{X}}^{(t-1)}_{\Delta,\text{corr}},]$. For every generated text FOL at iteration $(t)$, we collect the experience tuple $\langle \boldsymbol{x}'^{(t)}_{\text{FOL}}, \boldsymbol{x}_{\text{NL}}, \hat{\mathcal{X}}_{\Delta,\text{prev}}, \hat{\boldsymbol{x}}_{\text{FOL}}, r^{(t)} \rangle$ where $r^{(t)} = \text{RM}(\boldsymbol{x}_{\text{FOL}}, \boldsymbol{x}'^{(t)}_{\text{FOL}})$, and once enough experience is collected, we update parameters $\theta$ via PPO (Schulman et al., 2017). We summarize key aspects of four tasks in Appendix C.

## 4.3 FOL EVALUATION

Consider two FOL rules (denoted as $R$ and $R'$) "$R : \neg(P(A) \wedge P(B))$" and "$R' : \neg P(A) \vee \neg P(B)$" — $R$ and $R'$ are logically equivalent but are different in plain text; also consider a pair of rules "$R : \forall x P(x)$" and "$R' : \forall x \forall y P(x) \wedge Q(y)$"— if $R$ is the ground-truth and $R'$ the LLM prediction, how should one measure the distance between the two?

**Logical equivalence (LE)**. We propose to measure the logical equivalence between the rules by matching their truth tables and computing the overlap ratio. We introduce this with a running example in Figure 4. Specifically, let $R$ and $R'$ be the two rules parsed from the text $\boldsymbol{x}_{\text{FOL}}$ and $\boldsymbol{x}'_{\text{FOL}}$. We identify the set of literals in each rule $\mathcal{P} = [p_1, p_2, ...]$ and $\mathcal{Q} = [q_1, q_2, ...]$. In the case of Figure 4, $\mathcal{P} = [\texttt{Country}(x), \texttt{InEU}(x), \texttt{EUCountry}(x)]$ and $\mathcal{Q} = [\texttt{LocatedInEU}(y), \texttt{EUCountry}(y)]$. One can consider the set of literals as an array of Boolean variables, and the FOL as a circuit that takes in the Boolean values and outputs a single Boolean value. Therefore, we can represent a FOL with a truth table that enumerates all possible inputs and the resulting outputs. And to compare $R$ and $R'$, we count the number of configurations that match and divide it by the total number of configurations; this yields a score in $[0, 1]$. In Figure 4, this is $7/8 = 0.875$. The main issue with this approach is finding the right input bindings between $\mathcal{P}$ and $\mathcal{Q}$, and dealing with the case where the numbers of inputs are different (i.e., $|\mathcal{P}| \neq |\mathcal{Q}|$). We solve this by finding the binding that gives the highest LE score via greedy search and filling the rest of the missing inputs with dummy inputs. In Figure 4, $\texttt{InEU}(x)$ binds to $\texttt{LocatedInEU}(y)$ and $\texttt{EUCountry}(x)$ binds to $\texttt{EUCountry}(y)$; and we fill in a dummy in $\mathcal{Q}$ to match $\texttt{Country}(x)$ in $\mathcal{P}$. We leave more details in Appendix D.

**Reward design for RLHF correction**. The reward model RM in **(T4)** measures the similarity between two text FOLs $\boldsymbol{x}_{\text{FOL}}, \boldsymbol{x}'_{\text{FOL}}$. Therefore, we use the LE score as the main source of the reward. However, we also want the model to extract the right predicate and entity names from the NL statement. We incorporate this aspect by computing the BLEU score between the text $\boldsymbol{x}_{\text{FOL}}$ and $\boldsymbol{x}'_{\text{FOL}}$ with a specialized FOL tokenizer. We set the final reward as the mixture of the two: $\text{RM}(\boldsymbol{x}_{\text{FOL}}, \boldsymbol{x}'_{\text{FOL}}) = \omega * \text{LE}(R, R') + (1 - \omega) * \text{BLEU}(\boldsymbol{x}_{\text{FOL}}, \boldsymbol{x}'_{\text{FOL}})$, where $\omega$ is the mixing ratio and in experiments we set it to 0.7. With this setting, we encourage the model to prioritize on generating the logical equivalent rules rather than ones that are similar in plain text with a high BLEU score.

---

[4]Technically, **(T4)** does not involve human feedback, but we keep the name since the protocols are the same.

Table 2: BLEU and logical equivalence (LE) scores of LLMs. In all three benchmarks, LOGI-CLLAMA outperforms GPT-3.5, and achieves similar performance as 5-shot GPT-4 with iterative correction. We skip GPT-4 evaluation on MALLS as it is the model that generates it.

| Methods | LogicNLI | | FOLIO | | MALLS | |
|---|---|---|---|---|---|---|
| | FOL BLEU | FOL LE | FOL BLEU | FOL LE | FOL BLEU | FOL LE |
| LLaMA2-7B 5-shot | 0.697 | 0.783 | 0.288 | 0.654 | 0.459 | 0.506 |
| LLaMA2-13B 5-shot | 0.806 | 0.867 | 0.349 | 0.747 | 0.416 | 0.584 |
| GPT3.5 0-shot | 0.584 | 0.589 | 0.248 | 0.429 | 0.407 | 0.141 |
| GPT3.5 5-shot | 0.905 | 0.918 | 0.341 | 0.767 | 0.497 | 0.648 |
| GPT4 0-shot | 0.740 | 0.863 | 0.372 | 0.799 | - | - |
| GPT4 5-shot | 0.913 | 0.989 | 0.400 | 0.855 | - | - |
| LogicLLaMA-7B Trans. | 0.912 | 0.965 | 0.354 | 0.826 | 0.762 | 0.910 |
| LogicLLaMA-7B Corre. | 0.913 | 0.970 | 0.368 | 0.827 | 0.767 | 0.910 |
| LogicLLaMA-7B RLHF Corre. | 0.923 | 0.979 | 0.378 | 0.841 | 0.774 | 0.920 |
| LogicLLaMA-13B Trans. | 0.922 | 0.978 | 0.361 | 0.832 | 0.765 | 0.916 |
| LogicLLaMA-13B Corre. | 0.925 | 0.982 | 0.377 | 0.852 | 0.769 | 0.918 |
| LogicLLaMA-13B RLHF Corre. | 0.927 | 0.986 | 0.384 | 0.858 | 0.778 | 0.924 |

## 5 EXPERIMENTS

We address the following questions in the experiment section: (**Q1**) How good is MALLS? Can we train a strong NL-FOL translation model that also generalizes to other datasets such as FOLIO? (**Q2**) How well does the LOGICLLAMA perform in direct translation mode and correction mode? (**Q3**) How do the iterative corrections influence the performance of LOGICLLAMA?

**Dataset**. For MALLS, we use the human-annotated subset as the test set; we hold out 2K samples as the valid set, and the rest samples are used for training **(T1-4)**; we also include 1K pairs from the training set of LogicNLI since it has a different rule distribution where rules are mostly grounded rules (i.e., many of them do not contain any variables) instead of FOL rules. We evaluate the LLMs on MALLS test set as well as the full FOLIO dataset and the test set of LogicNLI.

**Training, generation, and hardware settings**. For all training tasks, we fine-tune LOGICLLAMA using LoRA with rank=8, $\alpha = 8$, and dropout 0.05 for all linear layers in the attention block and feedforward nets. We use the AdamW optimizer (Loshchilov & Hutter, 2017) with $lr = 0.0003$. For the generation, we use a cutoff length of 256 for **(T1-2)** and 1024 for **(T3-4)**, where 748 and 256 are allocated for the input prompt and output sequences respectively. All experiments are conducted on an i7-8700K machine with 32G RAM and a single 4090 GPU (Detailed settings at Appendix E).

**Metrics**. We evaluate the translated and the final corrected FOL rules with two metrics: FOL BLEU score and FOL logical equivalence (LE) score (§4.3).

### 5.1 RESULTS

Table 2 shows the results of LOGICLLAMA and GPT models on LogicNLI, FOLIO, and MALLS dataset. In general, we found that 5-shot GPT-4, as the most powerful LLM to date, achieves the best performance for both benchmarks (Note that we did not evaluate it on MALLS as it is the same model that generates it). On the other hand, LOGICLLAMA outperforms GPT-3.5 models in both translation and correction modes, and the best performance is achieved by RLHF correction which leads to a GPT-4 level performance. This suggests that MALLS can indeed produce an LLM comparable to GPT-4 on a gold set, which addresses the question (**Q1**).

Benchmark-wise, all methods achieve near-perfect results on LogicNLI except for 0-shot GPT-3.5, which has trouble generating syntactically valid rules due to the lack of examples. This is because LogicNLI is synthetically generated and the rules all share a similar FOL template. On the other hand, FOLIO is more challenging on the BLEU metric: meaning LLMs fail to recognize the right predicate names. This is largely due to FOLIO FOL rules usually omit details in the NL sentence and use highly simplified predicate names. Lastly, MALLS is more balanced in terms of BLEU and LE metrics and is generally most challenging for in-context learning among the three benchmarks.

### 5.2 ANALYSIS

**Translation vs. Correction**. Table 2 suggested that the correction mode LOGICLLAMA leads to better performance than the direct translation mode. This confirms our intuition in §4.1 and

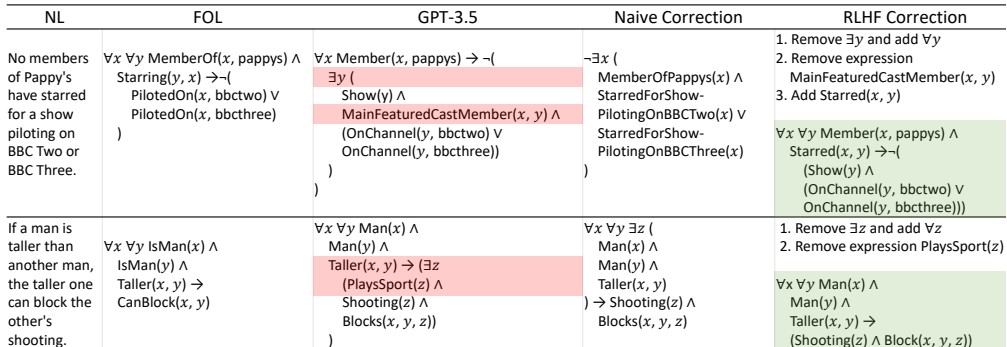

| NL | FOL | GPT-3.5 | Naive Correction | RLHF Correction |
|---|---|---|---|---|
| No members of Pappy's have starred for a show piloting on BBC Two or BBC Three. | $\forall x \, \forall y \, \text{MemberOf}(x, \text{pappys}) \land$ $\text{Starring}(y, x) \rightarrow \neg ($ $\text{PilotedOn}(x, \text{bbctwo}) \lor$ $\text{PilotedOn}(x, \text{bbcthree})$ $)$ | $\forall x \, \text{Member}(x, \text{pappys}) \rightarrow \neg ($ $\exists y \,($ $\text{Show}(y) \land$ $\text{MainFeaturedCastMember}(x, y) \land$ $(\text{OnChannel}(y, \text{bbctwo}) \lor$ $\text{OnChannel}(y, \text{bbcthree}))$ $)$ $)$ | $\neg \exists x \,($ $\text{MemberOfPappys}(x) \land$ $\text{StarredForShow-}$ $\text{PilotingOnBBCTwo}(x) \lor$ $\text{StarredForShow-}$ $\text{PilotingOnBBCThree}(x)$ $)$ | 1. Remove $\exists y$ and add $\forall y$ 2. Remove expression $\quad$ MainFeaturedCastMember$(x, y)$ 3. Add Starred$(x, y)$ 

 $\forall x \, \forall y \, \text{Member}(x, \text{pappys}) \land$ $\text{Starred}(x, y) \rightarrow \neg ($ $(\text{Show}(y) \land$ $(\text{OnChannel}(y, \text{bbctwo}) \lor$ $\text{OnChannel}(y, \text{bbcthree})))$ |
| If a man is taller than another man, the taller one can block the other's shooting. | $\forall x \, \forall y \, \text{IsMan}(x) \land$ $\text{IsMan}(y) \land$ $\text{Taller}(x, y) \rightarrow$ $\text{CanBlock}(x, y)$ | $\forall x \, \forall y \, \text{Man}(x) \land$ $\text{Man}(y) \land$ $\text{Taller}(x, y) \rightarrow (\exists z$ $(\text{PlaysSport}(z) \land$ $\text{Shooting}(z) \land$ $\text{Blocks}(x, y, z))$ $)$ | $\forall x \, \forall y \, \exists z \,($ $\text{Man}(x) \land$ $\text{Man}(y) \land$ $\text{Taller}(x, y)$ $) \rightarrow \text{Shooting}(z) \land$ $\text{Blocks}(x, y, z)$ | 1. Remove $\exists z$ and add $\forall z$ 2. Remove expression PlaysSport$(z)$ 

 $\forall x \, \forall y \, \text{Man}(x) \land$ $\text{Man}(y) \land$ $\text{Taller}(x, y) \rightarrow$ $(\text{Shooting}(z) \land \text{Block}(x, y, z))$ |

Figure 5: Examples of correcting GPT-3.5's output via naive and RLHF correction.

addresses the question (**Q2**). More importantly, these results suggest a new paradigm of future LLM development: by training a local LLM on the output of a more powerful model, one can conduct in-depth customization on the model behavior while still leveraging the generalizability of the powerful LLMs for heavy lifting. This paradigm is beneficial as GPT-3.5 and GPT-4 nowadays have limited support for customization.

Table 3: RLHF correction performance vs. Max # generations.

| Metrics | Max # Generations | | | |
|---|---|---|---|---|
| | 1 | 3 | 5 | 10 |
| FOL BLEU | 0.357 | 0.373 | 0.375 | 0.381 |
| FOL LE | 0.822 | 0.831 | 0.834 | 0.835 |

Figure 6: LOGICLLAMA-7B correction performance averaged over corresponding GPT-3.5 LE and BLEU scores.

**Effect of iterative correction**. To see how and why iterative correction improves performance, we compare the (**T2**) naive and the (**T4**) RLHF correction performance on samples grouped by their "difficulty" level. To do this, we group samples by the GPT-3.5's LE and BLEU scores into several bins (e.g., [1.0-0.9], [0.9-0.8] and etc.). Within each bin, we average the scores of GPT-3.5, (**T2**), and (**T4**). The results of LOGICLLAMA-7B on FOLIO are shown in Figure 6. And correction examples are shown in Figure 5. We find the RLHF correction leads to a better performance generally by improving the difficult examples where GPT-3.5 fails significantly. The same trend is also present between (**T2**) and (**T4**), where correction leads to better performance, especially on the BLEU score. We conjecture this is because the iterative steps make it easy to find the right predicate and entity names. Compared to the LE score, both LOGICLLAMA and GPT have a significant drop in BLEU score for difficult samples. This is because BLEU score is calculated on plain text and is more sensitive to different naming and ordering of literals that are in fact logically equivalent (as discussed in §4.3), especially in the long FOL rules.

**Effect of step length**. We study the effect of the steps by varying the maximum number of allowed generations on a single sample, which effectively limits the number of corrections that could be made by the model. The results of LOGICLLAMA-7B on FOLIO are shown in Table 3. We found the performance saturated quickly for a max of three generations. The one-generation case is slightly worse than the naive correction counterpart due to only a limited number of corrections being made.

## 6 CONCLUSION

We release MALLS, a high-quality dataset of 28K verified sentence-level NL-FOL pairs collected from GPT-4. During the verification process, we identify patterns of errors made by GPT-4. This underscores the difficulty in achieving accurate NL-FOL translation, warranting further research. We also present LOGICLLAMA, the first specialized LLMs for the NL-FOL translation task fine-tuned on MALLS. In experiments, LOGICLLAMA shows competitive performance with GPT-4, while outperforming GPT-3.5 on three NL-FOL benchmarks. Through an RLHF training framework, we equip LOGICLLAMA with iterative corrective capability, allowing it to consistently correct its own outputs, as well as outputs from GPT-3.5.

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
