

Figure 7: Top 200 frequent FOL term pairs in MALLS. Many terms are associated with a wide range of other terms, which suggests the rules are semantically and contextually diverse.

## A APPENDIX

In the following, we provide further details about the dataset creation, logical equivalence computation and experimental settings.

## B MALLS DATASET CREATION DETAILS

### B.1 DATA COLLECTION

**Prompt table**. Table 4 shows the prompt templates used for prompt generation.

### B.2 FOL PARSING AND VERIFICATION

**FOL CFG grammar**. We define the FOL with the following CFG grammar:

S -> F | Q F

Q -> QUANT VAR | QUANT VAR Q

F -> '¬' | '(' F ')' | '(' F ')' | F OP F | L

OP -> '⊕' | '∨' | '∧' | '→' | '↔'

L -> '¬' PRED '(' TERMS ')' | PRED '(' TERMS ')'

TERMS -> TERM | TERM ',' TERMS

TERM -> CONST | VAR

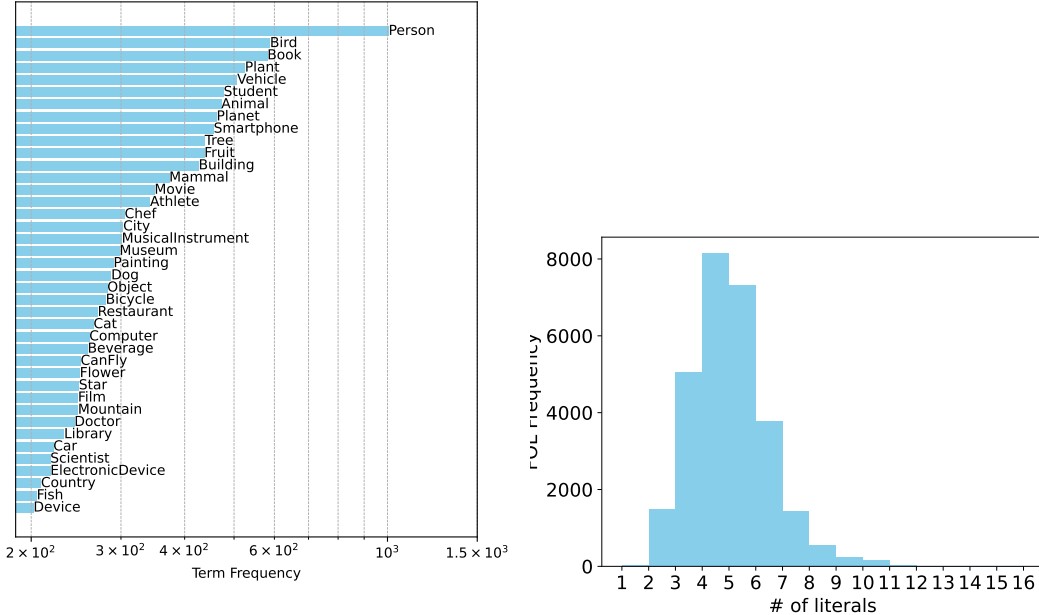

Figure 8: Top 40 frequent FOL terms (MALLS).   Figure 9: Literal frequency distribution (MALLS).

QUANT -> '∀' | '∃'

Note that, for `PRED`, `CONST`, and `VAR` they have corresponding production rules generated for each FOL rule example. For example, for rule "$\forall x((\texttt{Person}(x) \land \texttt{Drinks}(x)) \to \texttt{DependentOn}(x, \texttt{Caffeine}))$", the production rules are

$$\texttt{PRED} \to \texttt{"Person"} \mid \texttt{"Drinks"} \mid \texttt{"DependentOn"}$$
$$\texttt{CONST} \to \texttt{"Caffeine"}$$
$$\texttt{VAR} \to \texttt{"x"}.$$

We show two example parse trees in Figure 10 and Figure 11.

### B.3 NL-FOL ALIGNMENT CHECK

We recruit 6 graduate students to label 1K samples, where each sample is annotated by 3 annotators on a 3-point scale: correct ,partial, and incorrect. We provide detailed instruction as follows:

You will evaluate how well the FOL rule aligns with the meaning of the NL sentence on a 3-point scale: Correct, Partial, and Incorrect. Let's use a running example to show you how to rate the translation:

"""

NL: `A turtle has a shell and can swim.`

FOL: $\exists x(\texttt{Turtle}(x) \land \texttt{Shell}(x) \land \texttt{CanSwim}(x))$

"""

Your evaluation involves checking the following rubrics:

- Whether the rule has the right predicates, i.e., "Turtle", "Shell", and "CanSwim".
  - Having a slightly different predicate name is fine (e.g., "Shell" instead of "hasShell"); this is still a Correct example.
  - However, if some important notions in the NL are missing in the FOL (e.g., "Shell" missing in the FOL) or some predicates in FOL are not present in the NL (e.g., "CanRun(x)" found in FOL), it is one Partial error.

Table 4: List of prompt templates used for prompting GPT4 for NL-FOL pairs.

| | |
|---|---|
| System prompt | I want to create a dataset for translating natural language (NL) statements into first-order logic (FOL) rules. You will help me to create a diverse set of NL-FOL pairs.

For natural language (NL) generation, you should:
1. Come up with a statement stating either complex or simple real-world commonsense facts
2. The statements are meaningful, and diverse from each other

For FOL rule generation:
1. You SHOULD USE the following logical operators: $\oplus$ (either or), $\vee$ (disjunction), $\wedge$ (conjunction), $\rightarrow$ (implication), $\forall$ (universal), $\exists$ (existential), $\neg$ (negation), $\leftrightarrow$ (equivalence)
2. You *SHOULD NEVER USE* the following symbols for FOL: "'", "$\neq$", "%", "="
3. The literals in FOL SHOULD ALWAYS have predicate and entities, e.g., "Rounded(x, y)" or "City(guilin)"; expressions such as "y = a $\vee$ y = b" or "a $\wedge$ b $\wedge$ c" are NOT ALLOWED
4. The FOL rule SHOULD ACCURATELY reflect the meaning of the NL statement
5. You SHOULD ALWAYS put quantifiers and variables at the beginning of the FOL
6. You SHOULD generate FOL rules with either: (1) no variables; (2) one variable "x"; (3) two variables "x", "y"; or (4) three variables "x", "y" and "z"

Generation Format: you SHOULD ALWAYS generate the NL and FOL pairs in the following format
"""
— NL:
{your generated NL}
—
— FOL:
{your generated FOL}
—
""" |
| Initial FOLIO Few-shot examples prompt | — NL:
If someone is entire, then he is not serious, and vice versa.
— FOL:
$\exists x$ entire(x) $\leftrightarrow$ $\neg$serious(x)

— NL:
If there is at least one people who is both not excited and not timid, then Jonathan is elderly.
— FOL:
$\forall x$ ($\neg$excited(x) $\wedge$ $\neg$timid(x)) $\rightarrow$ elderly(Jonathan)

— NL:
Someone who is eithor not fresh or entire is always not serious.
— FOL:
$\forall x$ ($\neg$concerned(x) $\vee$ fresh(x)) $\rightarrow$ entire(John)

— NL:
If Nathalie is not blue, then Collier is entire.
— FOL:
$\neg$blue(Nathalie) $\rightarrow$ entire(Collier)

— NL:
Someone is courteous and not elderly if and only if he is not excited and not various.
— FOL:
$\exists x$ (courteous(x) $\wedge$ $\neg$elderly(x)) $\leftrightarrow$ ($\neg$excited(x) $\wedge$ $\neg$various(x)) |
| Negative N-gram prompt | They DO NOT involve concepts and terms (and the synonyms) such as "considered","person","either","water", "if it has","if it is","it has a","is considered a","A person is" |
| FOL prompts | They are [complex \| simple] statements involving at least [1 \| 2 \| 3] logical variables |
| | The statement involves diverse logical operators such as logical negation, logical xor and disjunction |
| Break-down prompt | [IMPORTANT] AVOID making long predicate names like "MoonShinesAtNight","SunShinesDuringDay" |

- Whether the rule has the right quantifiers.

    - You should only examine this item if the NL explicitly contains the following phrases: If NL explicitly says "Every A has sth" or "For all A..." then the FOL must have "$\forall$". On the other hand, if it says "Some A has sth" or "There exists A that...", then the FOL must have "$\exists$". Failing to have the right quantifiers is one Partial error.

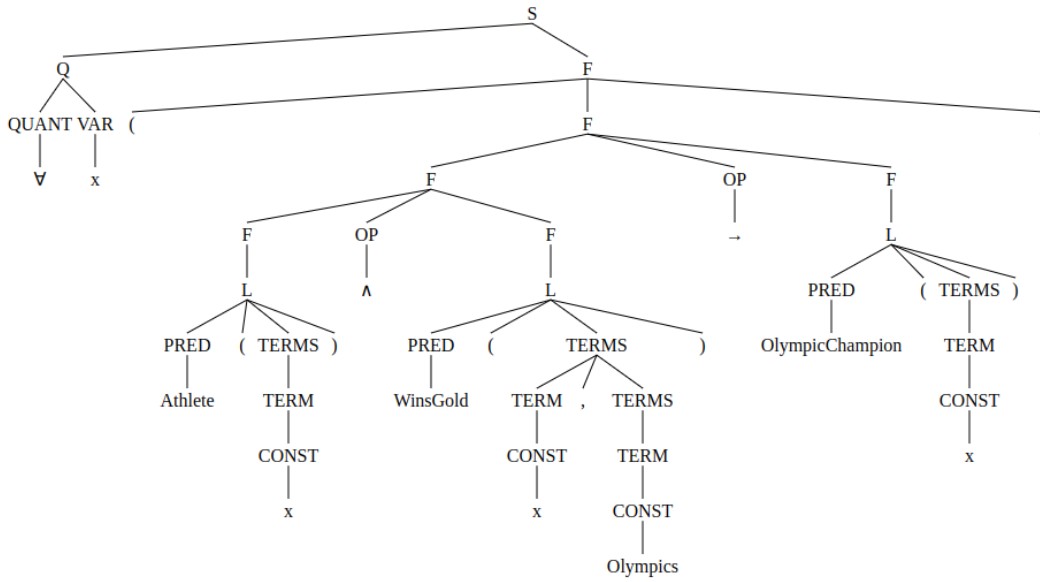

Figure 10: CFG parse tree of FOL rule $\forall x(\texttt{Athlete}(x) \wedge \texttt{WinsGold}(x,\texttt{Olympics}) \rightarrow \texttt{OlympicChampion}(x))$.

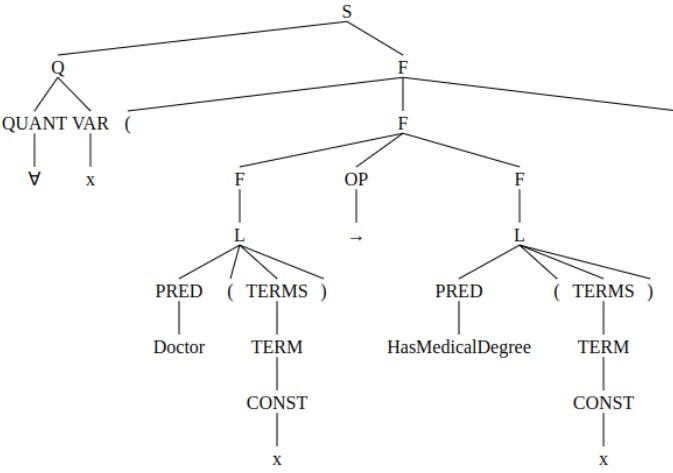

Figure 11: CFG parse tree of FOL rule $\forall x(\texttt{Doctor}(x) \rightarrow \texttt{HasMedicalDegree}(x))$.

- – If NL does not have these explicit hints, then using either "∀" or "∃" is fine (e.g., the turtle example above can use either "∀" or "∃")
- • Whether the FOL rule has the same logical meaning as the NL sentence. This involves comprehending the meaning FOL rule and comparing it against the NL sentence
  - – FOL misaligns with the NL meaning due to a wrong expression being used. It is one Partial error if the FOL simply misaligns with but not contradicts the NL; but if the meaning explicitly contradicts the NL, then it is an Incorrect example. E.g.:
    - ∗ Using the wrong logic operators, e.g., "Turtle(x) ∨ Shell(x) " instead of "Turtle(x) ∧ Shell(x)", is an Incorrect example as the meaning is obviously different.
    - ∗ Having the wrong implication, e.g., "A person has legs" translated into "HasLegs(x) → Person(X)", where it should be "Person(X) → HasLegs(x)". This is also an Incorrect example.
  - – Caveat on "→" vs "∧" vs "↔": Sometimes these three operators can be used interchangeably to express the same meaning despite they are logically inequivalent. For example,

* "Turtle(x) → Shell(x) ∧ CanSwim(x)", "Turtle(x) ∧ Shell(x) ∧ CanSwim(x)", and "Turtle(x) ↔ Shell(x) ∧ CanSwim(x)" all express a similar meaning, and should be considered a Correct case.
        * However, there are cases where they obviously lead to different meanings and this should be considered one Partial error.
    – Caveat on "∨" vs "⊕": we know "A ∨ B" includes the case that A and B are both True, whereas, "A ⊕ B" means either A or B is True but not both. So how to judge which one is better in the alignment?
        * If NL explicitly says "either A or B but not both" or something similar, then FOL must use "⊕", otherwise using "∨" is fine even if it does not reflect the reality. E.g., "water is solid or liquid" then it is fine to have "Water(x) ∨ Solid(x) ∨ liquid(x)" even though we know it cannot be in two states at the same time.
        * If NL does not explicitly say the above, but FOL contains "⊕", then we need to judge with common sense. For example, "Water(x) → Solid(x) ⊕ liquid(x)" also aligns with the above despite no explicit hints in NL.
    – Logic variable issues:
        * Associating the wrong logic variables, e.g.: "if x is greater than y and y is greater than z, then x is greater than z" translated to "GreaterThan(x, y) ∧ GreaterThan(y, z) → GreaterThan(z, x)", where it should be "GreaterThan(x, y) ∧ GreaterThan(y, z) → GreaterThan(x, z)". This is one Partial error.
        * Redefining variables, e.g. "Turtle(x) ∧ Fish(x) ∧ CanSwim(x)", where variable x is defined with two unary predicates (or classes) where one does not subsume the other one. This is one Partial error. For cases like "Animal(x) ∧ Turtle(x)", it is a Correct use because Animal subsumes Turtle
        * Free/redundant logic variables, e.g.:
            · "Person(x) ∧ Has(x, y)" — It has a free variable/redundant variable y, which is one Partial error.
            · "Person(x) ∧ Has(x, y) → Ownby(y, x)" — Note that this is a Correct use of variables: variables do not need to be defined by unary predicates (or "classes") to be valid, e.g., "Tool(y)", as long it appears at least twice in different literals.

By checking the above rubrics, you will rate the rule with one of the four ratings:

* Correct:
    – The FOL rule aligns with the meaning of the NL sentence
    – Possible slightly different predicate names or quantifiers.
    – Some ambiguities in the usage of "→" vs "∧" vs "↔", but it does not lead to an obviously different meaning
* Partial: one and only one of the following errors occur
    – The FOL rule does not align with the NL in some parts but does not explicitly contradict the NL
    – Missing important predicates, or having obviously irrelevant predicates
    – Wrong quantifier when the NL has explicit hints such as "For all…" or "Some…"
    – Wrong use of variables, such as free/redundant variables, wrong variable association, or redefining variables
* Incorrect:
    – More than one Partial error was found in an example
    – The FOL rule explicitly contradicts the NL meaning.

## B.4 MALLS STATISTICS

**General statistics**. Figure 8 and Figure 9 show the top 40 frequent FOL terms and the literal count distribution in MALLS.

**Frequent FOL term pairs**. Figure 7 shows the top 200 frequent FOL term pairs in MALLS.

Table 5: The list of all atomic perturbations.

| Operation Type | Subtypes | Original | Perturbed |
|---|---|---|---|
| Label Change | Change Predicate | $P(A) \wedge R(B)$ | $R(A) \wedge R(B)$ |
| | Change Term | $\forall x \; P(x) \wedge P(B)$ | $\forall y \; P(x) \wedge P(B)$ 
 $\forall x \; P(x) \wedge P(x)$ |
| | Change Operator | $\forall x \; P(x) \wedge P(B)$ | $\forall x \; P(x) \vee P(B)$ |
| Insert | Insert Term | $\forall x \; P(x) \wedge P(B)$ | $\forall x \, \exists y \; P(x) \wedge P(B)$ 
 $\forall x \; P(x) \wedge P(x, B)$ |
| | Insert Negation 
 Insert Formula | $P(A) \wedge P(B) \wedge P(C)$ 
 $P(A) \wedge P(B)$ | $P(A) \wedge \neg(P(B) \wedge P(C))$ 
 $P(A) \wedge P(B) \rightarrow R(C)$ |
| Delete | Delete Term | $\forall x \, \forall y \; P(x) \wedge R(x, y)$ 
 $\forall x \, \forall y \; P(x) \wedge R(x, y)$ | $\forall y \; P(x) \wedge R(x, y)$ 
 $\forall x \, \forall y \; P(x) \wedge R(y)$ |
| | Delete Negation 
 Delete Formula | $\neg(P(A) \wedge P(B))$ 
 $P(A) \wedge P(B) \wedge P(C)$ | $\neg(P(A) \wedge P(B))$ 
 $P(A) \wedge P(C)$ |

## C    DETAILS ON ITERATIVE CORRECTION VIA RLHF

### C.1    FOL RULE PERTURBATIONS AND SFT

Since we do not have the ground-truth steps $\hat{\mathcal{X}}_\Delta$ for the real output $\hat{x}_{\text{FOL}}$, we generate synthetic steps and the output sequence by randomly perturbing the FOL rules in MALLS.

We consider three types of atomic perturbations: *label change*, *insert*, and *delete*. As shown in Table 5, label change can be conducted on any terms or logic operators in a FOL rule; insert operation is applicable to term, negation, and formula; and delete operation can be considered as the inverse of insertion. Note that, we restrict the perturbations to only produce valid rules. The reasons are two-fold: (1) the invalid rule space is effectively the space of all possible strings which is prohibitive to explore; and (2) we found GPT-3.5 rarely generates syntactically invalid rule, thus, limiting the synthetic data in the valid rule space will already cover a wide range of the actual GPT-3.5 outputs.

**Perturbation process**. Given a ground-truth pair $\langle x_{\text{NL}}, x_{\text{FOL}} \rangle$, a parser (Appendix B) will parse $x_{\text{FOL}}$ into an abstract syntax tree (AST). We randomly perturb the AST with atomic operations in Table 5 and for $N_{\text{Perturb}}$ times. Here, $N_{\text{Perturb}}$ is also picked randomly from a list of numbers, and in the experiments, we set it to $\{0, 1, 2, ..., 10\}$. In the case $N_{\text{Perturb}} = 0$, the perturbed rule remains the same as the ground truth and the step is simply "No changes needed"; this is effectively a negative example that penalizes the model for over-correcting. During training, we found LOGICLLAMA still tends to over-correct the samples as negative samples by default account for around 10% of the data, so we manually set the probability of negative sample generation to 0.2.

**Iterative correction**. Depending on the capacity of the LLM, it might be difficult for the model to learn to output many steps (say 10) within one generation. We propose to break down the correction into multiple generations, where the model is tasked to output at most $N_{\text{Correct}}$ steps of correction given the perturbed rule and the previous corrections up to $N_{\text{Perturb}} - N_{\text{Correct}}$ steps. For example, the right column of Figure 2 shows an iterative correction sample: it requires total $N_{\text{Perturb}} = 4$ steps to correct the rule, but we picked a correction steps of $N_{\text{Correct}} = 2$; this means the perturbed rule together with the previous two steps are treated as input and the last two steps are the output. Similar to $N_{\text{Perturb}}$, we randomly choose $N_{\text{Correct}}$ from a list, where we set it to $\{0, 1, 2, 3\}$. And apparently, $N_{\text{Correct}}$ should be no greater than the total steps $N_{\text{Correct}} = \min(N_{\text{Correct}}, N_{\text{Perturb}})$.

**SFT correction**. For this **(T3)** task, we generate the synthetic dataset consisting of 150K examples in the form of $\langle x_{\text{NL}}, x_{\text{FOL}}, \hat{\mathcal{X}}_{\Delta,\text{prev}}, \hat{\mathcal{X}}_{\Delta,\text{corr}}, \hat{x}_{\text{FOL}} \rangle$ using the above method, where $\hat{\mathcal{X}}_{\Delta,\text{prev}}, \hat{\mathcal{X}}_{\Delta,\text{corr}}, \hat{x}_{\text{FOL}}$ are the previous correction steps, target correction steps and the perturbed FOL rule respectively. We then fine-tune the LLaMA-7B/13B with LoRA again using the standard autoregression objective: the input is $[x_{\text{NL}}, \hat{x}_{\text{FOL}}, \hat{\mathcal{X}}_{\Delta,\text{prev}}]$ and the output is $[\hat{\mathcal{X}}_{\Delta,\text{corr}}, x_{\text{FOL}}]$.

### C.2    SUMMARY OF TRAINING TASKS

We summarize the input, output, and the training objectives of **(T1-T4)** in Table 6.

Table 6: Summary of input, output and objectives of task **(T1-T4)**

| Task | Input | Output | Objective |
|---|---|---|---|
| **(T1)** Direction Translation | $\boldsymbol{x}_{\text{NL}}$ | $\boldsymbol{x}_{\text{FOL}}$ | Eq.(2) |
| **(T2)** Naive Correction | $[\boldsymbol{x}_{\text{NL}}, \hat{\boldsymbol{x}}_{\text{FOL}}]$ | $\boldsymbol{x}_{\text{FOL}}$ | Eq.(2) |
| **(T3)** SFT Correction | $[\boldsymbol{x}_{\text{NL}}, \hat{\boldsymbol{x}}_{\text{FOL}}, \hat{\mathcal{X}}_{\Delta,\text{prev}}]$ | $[\hat{\mathcal{X}}_{\Delta,\text{corr}}, \boldsymbol{x}_{\text{FOL}}]$ | Eq.(2) |
| **(T4)** RLHF Correction | $[\boldsymbol{x}_{\text{NL}}, \hat{\boldsymbol{x}}_{\text{FOL}}, \hat{\mathcal{X}}_{\Delta,\text{prev}}]$ | $[\hat{\mathcal{X}}_{\Delta,\text{corr}}, \boldsymbol{x}_{\text{FOL}}]$ | Eq.(3) |

For task **(T1-T3)**, a standard autoregressive loss is used: where the model is trained to predict the token $x_t$ given the previous tokens $\boldsymbol{x}_{<t}$ with cross-entropy loss

$$L(\theta) = \mathbb{E}_{x,t}[-\log p_\theta(x_{t,\text{out}}|\boldsymbol{x}_{<t,\text{out}}, \boldsymbol{x}_{\text{in}})]. \tag{2}$$

For **(T1-T3)**, the $\boldsymbol{x}_{\text{in}}$ and $\boldsymbol{x}_{\text{out}}$ are shown in 6.

For task **(T4)**, the RLHF objective is used: we implement a reward model $\text{RM}(\boldsymbol{x}_{\text{FOL}}, \boldsymbol{x}'_{\text{FOL}})$ that measures the logical equivalence between two FOL rules and train the model to maximize the reward with the PPO algorithm (Schulman et al., 2017). Specifically, the full reward is computed as

$$R(\boldsymbol{x}_{\text{FOL}}, \boldsymbol{x}'_{\text{FOL}}) = \text{RM}(\boldsymbol{x}_{\text{FOL}}, \boldsymbol{x}'_{\text{FOL}}) - \beta * \log \frac{\pi_\theta(\boldsymbol{x}_{\text{FOL}}, \hat{\mathcal{X}}_{\Delta,\text{corr}}|\boldsymbol{x}_{\text{NL}}, \hat{\mathcal{X}}_{\Delta,\text{prev}}, \hat{\boldsymbol{x}}_{\text{FOL}})}{\pi^{\text{ref}}(\boldsymbol{x}_{\text{FOL}}, \hat{\mathcal{X}}_{\Delta,\text{corr}}|\boldsymbol{x}_{\text{NL}}, \hat{\mathcal{X}}_{\Delta,\text{prev}}, \hat{\boldsymbol{x}}_{\text{FOL}})}, \tag{3}$$

where the policy $\pi_\theta$ is the language model $f_\theta$ with learnable parameters $\theta$ and $\pi^{\text{ref}}$ is the reference model from **(T3)** with parameters frozen. The second term in Eq.(3) is the KL-divergence between the learned policy and its original policy from **(T3)** and its strength is controlled by a hyperparameter $\beta$.

## D    COMPUTING LOGICAL EQUIVALENCE AND BLEU SCORE

**logical equivalence**. To train and evaluate LOGICLLAMA, we compute the logical equivalence score (LE) that measures the similarity between two rules $R$ and $R'$. The computation is done in three steps: (1) finding the literals of $R$ and $R'$, that is $\mathcal{P} = [p_1, p_2, ...]$ and $\mathcal{Q} = [q_1, q_2, ...]$; (2) binding the literals in $\mathcal{P}$ to those in $\mathcal{Q}$ (or vice versa); and (3) generating the truth tables for the binding and computing the score.

Finding the literals of a FOL rule is straightforward after we parse it into a CFG tree: we extract all the subtrees whose root label is L and remove possible duplicate literals. In the case where the parsing fails, we simply skip the rest of the computation and return a score of zero, as that indicates the rule is syntactically invalid.

The main challenge here is to determine the literal binding between $\mathcal{P}$ and $\mathcal{Q}$. Using Figure 4 as the example, $R$ has literals $\mathcal{P} = [\texttt{Country}(x), \texttt{InEU}(x), \texttt{EUCountry}(x)]$ and $R'$ has literals $\mathcal{Q} = [\texttt{LocatedInEU}(y), \texttt{EUCountry}(y)]$. We want to find the one-one matching for each of the literals, such that we can compare the truth tables. First, we address the case where $|\mathcal{P}| \neq |\mathcal{Q}|$ by adding DUMMY inputs to the shorter one, and in this example, it is $\mathcal{Q}$ which becomes $[\texttt{LocatedInEU}(y), \texttt{EUCountry}(y), \texttt{DUMMY1}]$. To match the literals, we first determine the matching strategy. Note that there are in total $!|\mathcal{Q}|$ numbers of bindings (permute $\mathcal{Q}$ when keeping $\mathcal{P}$) and there are many strategies to measure the match: for example, one can enumerate all bindings and compute the "average" score of all bindings or finding the worst case of the binding. Here, we choose to find the binding that yields the highest LE score, that is the "best" case binding. To do this, we implement a simplistic greedy search algorithm that iterates over each literal in $\mathcal{P}$ and finds the closest literal in $\mathcal{Q}$ in terms of edit distance. To avoid exponential numbers of bindings, we limit the search depth to 1000. Finally, given a binding between $\mathcal{P}$ and $\mathcal{Q}$, we compute the LE score by comparing the rows in their truth tables as the one shown in Figure 4.

**FOL BLEU score**. We use a specialized tokenizer for computing the FOL BLEU score. This tokenizer splits every quantifier, operator, and term into tokens. The split token sequence is the same as the leave nodes in the CFG parse tree (Figure 10 and Figure 11) listed in pre-order.

# E    EXPERIMENTAL SETTINGS

For all training tasks, we fine-tune LOGICLLAMA using LoRA with rank=8, $\alpha = 8$, and dropout 0.05 on all the linear layers in decoder blocks "[q_proj,k_proj,v_proj,o_proj, gate_proj,down_proj,up_proj]". We use the AdamW optimizer (Loshchilov & Hutter, 2017) with $lr = 0.0003$. For the generation, we use a cutoff length of 256 for **(T1)** and **(T2)**; and 1024 for **(T3)** and **(T4)**, where 748 and 256 are allocated for the input prompt and output sequences respectively. For **(T1-T3)**, the generation uses temperature=0.1, top_p=0.75, top_k=40 and num_beams=1. For **(T4)**, we adopt the setting suggested in the TRL library, which uses top_k = 0.0, top_p = 1.0, do_sample = True and no eos token; this effectively lets the model sample tokens from the logits and always generate to the full length. This generation configuration is needed to compute a valid KL divergence score between the actor model and the reference model (a copy of the same model before training).

Recall that **(T4)** generates corrections in multiple rounds of generations, where previous corrections are appended to the initial prompt and fed to the model again as the input prompt (Figure 2). For all experiments, we set the max rounds of generation to 10, except for Table 3 which examines the model's performance by varying the max rounds. Also, we found that GPT-3.5 can sometimes generate syntactically invalid FOL rules that lie outside of rule space simulated in **(T3)**. We address this by first correcting the GPT-3.5 response with naive correction **(T2)** and then feeding the output to **(T4)**.