# OpenReview forum: "Harnessing the Power of Large Language Models for Natural Language to First-Order Logic Translation"
_ICLR.cc/2024/Conference — Submitted to ICLR 2024_

### Official Review · Reviewer_8JQp · 2023-10-30

**Soundness:** 2 fair
**Presentation:** 2 fair
**Contribution:** 2 fair
**Rating:** 6
**Confidence:** 4

**Summary:**

This paper 28K sentence-level natural language to First Order Logic pairs collected from GPT-4. The authors also present a LLaMA-13B model fine-tuned on this dataset which combined with GPT3.5 achieves GPT-4 level performance on the NL-FOL translation tasks.

The combination method with GPT3.5 is interesting because it relies on using RLHF method to correct synthetically perturbed NL-FOL pairs and using a first order logic verifier as a reward model.

**Strengths:**

This paper presents a new method for gathering RLHF feedback based on iterative correction which is a non-trivial extension of previous methods.

**Weaknesses:**

The techniques in the paper are not novel and fine-tuning LLAMA for a downstream task can be considered engineering at this point instead of active research. The novelty of the proposed approach is not totally clear.

**Questions:**

Even though the paper says that the LogicLLAMA is finetuned using RLHF there is no human feedback involved. It should probably be called something else ?

---

> ### Author Response · Authors · 2023-11-16
> **Response**
>
> Thank you for your comments and we appreciate your positive feedback on our work. Our responses are as follows:
>
>
> ***”The novelty of the proposed approach is not totally clear”***
>
> We thank the reviewer for acknowledging the novelty of our proposed RLHF method. We would like to further highlight two important aspects of our work to clarify the overall novelty:
>
> **LogicLLaMA and MALLS are significant contributions to the logic and NLP literature.**
> NL-FOL translation has been a long-standing challenge in both NLP and the formal logic literature, and it plays a central role in many logic-based AI systems [1-4].
> Solving this task could open up a wide range of applications, yet there lacks such a translation model that scales to real-world data, preventing these systems from applying to real-world NLP problems.
>
> We believe LogicLLaMA is a significant contribution to the literature (As also acknowledged by reviewer wVqn) as it, for the first time, provides a GPT4-level performance translation model that is cheap, open-source, and can be extended for downstream tasks.
>
>
>
> **The creation of MALLS dataset is non-trivial; prior work with similar scope is widely appreciated.**
> This work shares a similar scope as those self-instruct LLM work such as Alpaca [5], Vicuna [6], and LLaVa [7]: most of them use the existing models and training algorithms, and the main contributions are “engineering” work such as dataset creation, prompting techniques, and so on, yet they are highly influential and significant to the community.
>
> We note the creation of the MALLS dataset is also one of the main contributions of this work and should not be neglected. Given the importance of NL-FOL translation in many logic-based NLP systems, a high-quality real-world NL-FOL pair dataset is highly valuable to the literature and can open up possibilities for many downstream applications. In this regard, MALLS stands out as it is significantly larger (14x times compared to FOLIO) and more diverse in terms of logical expression and context.
>
>
>
> **RLHF naming**
>
> Sorry for the confusion. Yes, there is indeed no human feedback involved. Technically, it should be RL with FOL verifier feedback, but since the overall algorithm remains the same we keep the same naming convention. We have included clarification for this in the draft.
>
>
> [1] Abzianidze, L. (2017). LangPro: Natural Language Theorem Prover. Proceedings of the 2017 Conference on Empirical Methods in Natural           Language Processing: System Demonstrations, 115–120.
>
> [2] Bos, J., & Markert, K. (2005). Recognising Textual Entailment with Logical Inference. Proceedings of Human Language Technology Conference and Conference on Empirical Methods in Natural Language Processing, 628–635.
>
> [3] Lu, X., West, P., Zellers, R., Bras, R. L., Bhagavatula, C., & Choi, Y. (2021). NeuroLogic Decoding: (Un)supervised Neural Text Generation with Predicate Logic Constraints (arXiv:2010.12884).
>
> [4] Wang, S., Zhong, W., Tang, D., Wei, Z., Fan, Z., Jiang, D., Zhou, M., & Duan, N. (2021). Logic-Driven Context Extension and Data Augmentation for Logical Reasoning of Text (arXiv:2105.03659).
>
> [5] Rohan Taori, Ishaan Gulrajani, Tianyi Zhang, Yann Dubois, Xuechen Li, Carlos Guestrin, Percy Liang, and Tatsunori B. Hashimoto. Stanford alpaca: An instruction-following llama model.
>
> [6] Wei-Lin Chiang, Zhuohan Li, Zi Lin, Ying Sheng, Zhanghao Wu, Hao Zhang, Lianmin Zheng,Siyuan Zhuang, Yonghao Zhuang, Joseph E. Gonzalez, Ion Stoica, and Eric P. Xing. Vicuna: An open-source chatbot impressing gpt-4 with 90%* chatgpt quality, March 2023.
>
> [7] Liu, Haotian, et al. "Visual instruction tuning." arXiv preprint arXiv:2304.08485 (2023).

---

### Official Review · Reviewer_6TG1 · 2023-10-31

**Soundness:** 3 good
**Presentation:** 2 fair
**Contribution:** 2 fair
**Rating:** 3
**Confidence:** 4

**Summary:**

The paper aims to address the challenge of translating natural language sentences into first-order logic (FOL). A new dataset, MALLS, is proposed, which is generated by leveraging LLMs and applying filtering rules to eliminate bad cases. The paper proposed fine-tuning LLaMA on MALLS to enhance LLaMA's FOL generation ability. The model is tested on LogicNLI, FOLIO, and MALLS to demonstrate the effectiveness of the proposed method.

**Strengths:**

The paper demonstrates considerable effort in conducting numerous experiments to achieve its objective.
The motivation to enhance FOL generation ability is interesting.

**Weaknesses:**

1. The paper lacks novelty. Most contributions are either engineering work or published work including: prompting LLM to generate data, various prompting and dataset filtering methods, supervised fine-tuning, and RLHF. Moreover, the conclusions are predictable: fine-tuning LLM on a specific domain can improve its performance, potentially surpassing the best LLM for open domains.
2. The proposed methods do not show significant improvement. As per Table 2, the majority of the gain comes from vanilla supervised fine-tuning on MALLS, and the newly proposed methods only bring marginal gains, especially when compared to the gain from the baseline to vanilla fine-tuning.

**Questions:**

1. The abstract should be written in a single paragraph.
2. Consider reducing the use of markers like C1-C2, Q1-Q2, T1-T4. The frequent use of these symbols indicates a struggle to explain clearly, and it can be challenging for readers to understand by locating the definition. Short names might be more effective.
3. Try to limit the number of your contributions and new terms. Highlight only the most significant things you want readers to remember. Currently, you have five prompting methods to create the dataset, four rules to filter the dataset, four fine-tuning methods to train LLaMA, and the experiment aims to address four research questions. This approach dilutes each contribution and makes it difficult to determine whether it works and can be generalized to new problems.
4. Can your method generate more examples, since your generation and filtering are automatically executed?
5. How do you assess the coverage or diversity of your dataset? LLM may only generate high-frequency knowledge.
6. While the natural language input generated from LLM is often fluent and grammatically correct, real human inputs can be noisy. How do you address this issue?

---

> ### Author Response · Authors · 2023-11-16
> **Response**
>
> Thank you for your comments. Our responses are as follows:
>
> &nbsp;
>
> ***”The paper lacks novelty”***
>
> We clarify this point on three aspects:
>
>
> **The iterative correction with RLHF is technically novel.**
> We emphasize that the iterative correction with RLHF is a novel extension (as also acknowledged by reviewer 8JQp) to the original RLHF setup. Prior work either realized iterative correction via pure prompting [1] with closed-source models such as ChatGPT, or utilized RLHF for other tasks [2] such as instruction-following fine-tuning. Ours is the first method that uses RLHF to fine-tune language models for logical understanding with non-trivial settings, such as FOL rule perturbation and logical equivalence evaluation.
>
>
> **LogicLLaMA and MALLS are significant contributions to the logic and NLP literature.**
> NL-FOL translation has been a long-standing challenge in both NLP and the formal logic literature, and it plays a central role in many logic-based AI systems [3-6].
> Solving this task could open up a wide range of applications, yet there lacks such a translation model that scales to real-world data, preventing these systems from applying to real-world NLP problems.
>
> We believe LogicLLaMA is a significant contribution to the literature (As also acknowledged by reviewer wVqn) as it, for the first time, provides a GPT4-level performance translation model that is cheap, open-source, and can be extended for downstream tasks.
>
>
>
> **The creation of MALLS dataset is non-trivial; prior work with similar scope is widely appreciated.**
> This work shares a similar scope as those self-instruct LLM work such as Alpaca [7], Vicuna [8], and LLaVa [9]: most of them use the existing models and training algorithms, and the main contributions are those so-called “engineering work”: dataset creation, prompting techniques, and so on, yet they are highly influential and significant to the community.
>
> We emphasize the creation of the MALLS dataset is also one of the main contributions of this work and should not be neglected. Given the importance of NL-FOL translation in many logic-based NLP systems, a high-quality real-world NL-FOL pair dataset is highly valuable to the literature (for benchmarking or fine-tuning) and can open up possibilities for many downstream applications. In this regard, MALLS stands out as it is significantly larger (14x times compared to FOLIO) and more diverse in terms of logical expression and context.
>
> &nbsp;
>
> ***” The proposed methods do not show significant improvement”***
>
> As stated above, we disagree that one views RLHF as the sole proposed method and the rest two modes as the baselines—they are different modes of a LLaMA model proposed in this work for NL-FOL translation, and they are trained and evaluated on a dataset is non-trivial to collect and process.
>
> Nevertheless, we are glad to share that we have resolved a few implementational issues with RLHF and rerun the experiments. We find the updated iterative correction performs significantly better compared to the two other modes, leading to ~10% increase in both FOL BLEU and FOL LE on FOLIO and MALLS (whereas scores for  LogicNLI are already saturated). We have updated the RLHF scores in Table 2.
>
> &nbsp;
>
> ***” Can your method generate more examples?”***
>
> Yes, and we plan to collect more FOL rules in future work.
>
> &nbsp;
>
> ***” How do you assess the coverage or diversity of your dataset? LLM may only generate high-frequency knowledge.”***
>
> We believe the **Pair diversity** paragraph in section 3.3 has sufficiently addressed this concern. During the data collection phase, we implemented several prompting techniques to encourage rule and term diversity. As a result, *“MALLS has a total term vocabulary size of 49394 and the most frequent terms occur less than 2K times (Figure 8 in Appendix B), suggesting a diverse vocabulary distribution.”*. We also show in Table 1, Figure 1, and 7 that MALLS covers a wide range of concepts as well.
>
> &nbsp;
>
> ***”While the natural language input generated from LLM is often fluent and grammatically correct, real human inputs can be noisy. How do you address this issue?”***
>
> As stated in Table 1 and Section 2, FOLIO consists of NL-FOL pairs written by humans, and we believe evaluating LogicLLaMA on this benchmark shows how well the model performs with *real human inputs*. As a result, Table 2 suggests that LogicLLaMA performs well on it.

---

> > ### Author Response · Authors · 2023-11-16
> > **References**
> >
> > [1] Pan, L., Albalak, A., Wang, X., & Wang, W. Y. (2023). Logic-LM: Empowering Large Language Models with Symbolic Solvers for Faithful Logical Reasoning (arXiv:2305.12295). arXiv; 1,3.
> >
> > [2] Ouyang, Long, et al. "Training language models to follow instructions with human feedback." Advances in Neural Information Processing Systems 35 (2022): 27730-27744.
> >
> > [3] Abzianidze, L. (2017). LangPro: Natural Language Theorem Prover. Proceedings of the 2017 Conference on Empirical Methods in Natural Language Processing: System Demonstrations, 115–120.
> >
> > [4] Bos, J., & Markert, K. (2005). Recognising Textual Entailment with Logical Inference. Proceedings of Human Language Technology Conference and Conference on Empirical Methods in Natural Language Processing, 628–635.
> >
> > [5] Lu, X., West, P., Zellers, R., Bras, R. L., Bhagavatula, C., & Choi, Y. (2021). NeuroLogic Decoding: (Un)supervised Neural Text Generation with Predicate Logic Constraints (arXiv:2010.12884).
> >
> > [6] Wang, S., Zhong, W., Tang, D., Wei, Z., Fan, Z., Jiang, D., Zhou, M., & Duan, N. (2021). Logic-Driven Context Extension and Data Augmentation for Logical Reasoning of Text (arXiv:2105.03659).
> >
> > [7] Rohan Taori, Ishaan Gulrajani, Tianyi Zhang, Yann Dubois, Xuechen Li, Carlos Guestrin, Percy Liang, and Tatsunori B. Hashimoto. Stanford alpaca: An instruction-following llama model.
> >
> > [8] Wei-Lin Chiang, Zhuohan Li, and et al. Vicuna: An open-source chatbot impressing gpt-4 with 90%* chatgpt quality, March 2023.
> >
> > [9] Liu, Haotian, et al. "Visual instruction tuning." arXiv preprint arXiv:2304.08485 (2023).

---

> ### Comment · Reviewer_6TG1 · 2023-11-20
> **Concerns about the novelty**
>
> I agree with the author regarding the value of the MALLS dataset. However, I have some concerns about the contributions:
>
> 1. LogicLLaMA: Unlike Alpaca, Cicuna, and LLaVA, which aim to enhance open-domain instruction following ability or multi-modal ability, LogicLLaMA primarily focuses on first-order generation ability. I am concerned that this gain may result from adapting open-domain to a specific domain, potentially harming open-domain ability. To address this concern, could the author provide tests of LogicLLaMA on tasks beyond logic translation? At a minimum, please test the model on logical, reasoning, and math-related datasets, which may benefit from the ability to generate logic expressions.
>
> 2. Performance of RLFH correlation: Based on the updated Table 2, the gain of RLHF correlation is still marginal. I agree with the author that the "LogicLLaMA-13B Trans." setting achieves more than a 10% gain compared to "LLaMA2-13B 5-shot", demonstrating that fine-tuning with MALLS can improve performance on these three datasets. However, when evaluating the proposed methods "Corre" and "RLHF Corre", we should compare them with the "trans" setting. Most of the gains are less than 1%, which is marginal. To support my concern, could the author run the fine-tuning experiment multiple times and report the variance of the results?
>
> 3. Creation method of MALLS: Does the creation method possess novelty that could benefit the community and generalized to creation of other datasets? I am unable to determine this on my own.

---

> > ### Author Response · Authors · 2023-11-22
> > **Response**
> >
> > Thank you for your response. We are glad to see that the response has addressed some of your concerns. Our responses to your remaining concerns are as follows:
> >
> >  &nbsp;
> >
> > **Is LogicLLaMA intended to be used as a standalone model for open-domain problems? — NO**
> >
> > We draw the analogy to self-instruct methods not to emphasize the end use of the model (i.e., whether the model is used in the open domain or a specific domain) but the **methodology and process involved for creating such a model, and that the so-called “engineering work” is non-trivial in these work and so is in LogicLLaMA**.
> >
> > We also disagree that testing LogicLLaMA as a standalone model on some datasets could benefit this work, given that LogicLLaMA is specifically trained for the NL-FOL translation task.
> >
> > We re-emphasize that NL-FOL translation is a critical task in the logic-based NLP system, and being able to train such a model that achieves SOTA performance on this task is a significant contribution to the field. In the big picture, LogicLLaMA is intended to be used as a translator module together with downstream reasoning modules as those in [1, 3-6]. **Therefore, whether or not the model has lost capability as an open-domain standalone model is irrelevant to this work**.
> >
> > &nbsp;
> >
> > **RLHF performance**
> >
> > We re-emphasize that this work concerns creating a dataset and training a model, i.e., LogicLLaMA for NL-FOL translation, the iterative correction is one of the three modes proposed together in this work that gives an extra edge over the other two, and one should not consider RLHF the sole method proposed here.
> >
> > We show that iterative correction consistently leads to better performance by conducting the Wilcoxon signed-rank test between the results of iterative correction and naive correction. We show the p-values of FOL BLEU and FOL LE for FOLIO and MALLS (LogicNLI is omitted as the performance is saturated): For the 7B model, the p-values are 0.106, 0.041, 0.113, 0.053 and for the 13B model, the p-values are 0.039, 0.127, 0.151, 0.003 respectively. This means iterative correction is statistically significantly better than the naive correction.
> >
> >  &nbsp;
> >
> > **Novelty in dataset creation method**
> >
> > First of all, we re-emphasize that the creation of the MALLS dataset is itself a major contribution to the logic and NLP community and should not be neglected. While the utilization of LLMs to produce synthetic data is a widely employed technique, the automatic validation and filtration of this data present various non-trivial challenges, as elucidated in Section 3 and Appendix B in full details. We not only implemented a rigorous process but also carried out human evaluations to ensure that MALLS maintains a high standard of quality and encompasses various intricate challenges of this task.
> >
> > On top of this, we plan to release the data collection pipeline upon publication and believe this can also generalize to creating other logic-based datasets as well. Specifically, the FOL parser and verifier can be used for validating any FOL rules in text, making it easy to create NLI datasets such as FOLIO [10]; the rule perturbation method is novel and can be used for logic benchmarks with fine-grained evaluation on LLMs’ logical understanding; and, the FOL rule evaluation metrics can be used as supervisions in many downstream reasoning tasks such as that in Logic-LM [1].
> >
> >  &nbsp;
> >
> > [10] Han, S., Schoelkopf, H., Zhao, Y., Qi, Z., Riddell, M., Benson, L., Sun, L., Zubova, E., Qiao, Y., Burtell, M., Peng, D., Fan, J., Liu, Y., Wong, B., Sailor, M., Ni, A., Nan, L., Kasai, J., Yu, T., … Radev, D. (2022). FOLIO: Natural Language Reasoning with First-Order Logic (arXiv:2209.00840). arXiv. http://arxiv.org/abs/2209.00840

---

### Official Review · Reviewer_wVqn · 2023-11-05

**Soundness:** 3 good
**Presentation:** 3 good
**Contribution:** 3 good
**Rating:** 8
**Confidence:** 4

**Summary:**

This paper contributed the MALLS dataset and the LogicLLAMA models for natural language to first-order-logic translation tasks. The authors also describe a multi-stage training paradigm using closed-source GPT models to generate datasets for training open-source LLAM for the NL-FOL translation tasks. Experiments are conducted to demonstrate the effectiveness of the proposal.

**Strengths:**

• Investing the possible links between large language models and logical reasoning is an important topics to advance the methodology for AI.
• The paradigm of training open-sourced NL-FOL from open-sourced LLMs using data sourcing from close-sourced LLMs is a reasonable way to create NL-FOL models.
• Experiments demonstrate almost start-of-the-art performance of NL-FOL tasks with the proposed LogicLLAMA model from LLAMA.

**Weaknesses:**

• The targeting FOL language lacks a formal characterization. Is it an fully expressive First-order logic language or just the logic program subset?
•  The description of iterative correction via RLHF is difficult to follow lacking a few explicit description to task T4. In particular, it took me multiple-pass to understand why T4 is different from T3. It might worth a few rewriting to make this more explicit and easy to follow regarding the format of training data and loss functions.

**Questions:**

1. Page 7, regarding logical equivalence, which exact semantic models of FOL do you refer to? It is better to explicitly name the exact semantic models, such as Herbrand structure/universe if there is any; otherwise there might be ambiguity rendering the FOL being just a subset of FOL --- even if only a subset of FOL is supported it is already meaningful but please be specific.
2.  Please formally define the 4 training tasks T1-T4 with explicit input-output definitions and loss definitions (if not enough space, putting them into supplemental materials is acceptable).

---

> ### Author Response · Authors · 2023-11-16
> **Response**
>
> Thank you for your comments. We appreciate your positive feedback on our work! Our responses are as follows:
>
>
> ***”Is it a fully expressive First-order logic language or just the logic program subset?”***
>
> Yes, both MALLS dataset and LogicLLaMA are for general FOL rules. When collecting and validating the FOL rules, we do not enforce specific semantic models such as the Herbrand structure. On the logic expression level, the dataset does have some biases, where it contains more entailment-type FOL rules and fewer rules with rare logical operators such as xor. For measuring logical equivalence, we follow a similar protocol as in FOLIO[1] that converts them to propositional rules and compares the truth table.
>
>
>
> **Iterative correction presentation; input, output, and loss functions of T1-T4**
>
> Thank you for the suggestions. We have included a summary of these items in Appendix C, where we summarize the input, output, and objectives of the four tasks.
>
>
> [1] Han, S., Schoelkopf, H., Zhao, Y., Qi, Z., Riddell, M., Benson, L., Sun, L., Zubova, E., Qiao, Y., Burtell, M., Peng, D., Fan, J., Liu, Y., Wong, B., Sailor, M., Ni, A., Nan, L., Kasai, J., Yu, T., … Radev, D. (2022). FOLIO: Natural Language Reasoning with First-Order Logic (arXiv:2209.00840). arXiv. http://arxiv.org/abs/2209.00840

---

> > ### Comment · Reviewer_wVqn · 2023-11-23
> >
> > Thank the authors for making the efforts to response to my questions. After reading response, the nature of the targeting first-order logic still needs a few clarifity (e.g. whether it only supports finite domain predicates or infinite domains). Regarding the novelty, please address the concerns of other reviewers.

---

### Meta-Review · Area_Chair_vE8f · 2023-12-13

**Metareview:**

This paper introduces the MALLS dataset of NL-FOL pairs and finetunes LLaMA (LogicLLAMA) for the translation tasks. Apart from the dataset, the contributions seem weak. Also, the reviewers raised concerns regarding the lack of generalization of LogicLLaMA.

**Justification For Why Not Higher Score:**

NA

**Justification For Why Not Lower Score:**

NA

---

### Decision · Program_Chairs · 2024-01-16

Reject